# Experimental and Numerical Analysis of Low-Velocity Impact of Carbon Fibre-Based Non-Crimp Fabric Reinforced Thermoplastic Composites

**DOI:** 10.3390/polym13213642

**Published:** 2021-10-22

**Authors:** Muhammad Ameerul Atrash Mohsin, Lorenzo Iannucci, Emile S. Greenhalgh

**Affiliations:** 1Department of Aeronautics, Imperial College London, Exhibition Road, London SW7 2AZ, UK; lo.iannucci@imperial.ac.uk (L.I.); e.greenhalgh@imperial.ac.uk (E.S.G.); 2Empa, Swiss Federal Laboratories for Materials Science and Technology, Überland Str. 129, 8600 Dübendorf, Switzerland

**Keywords:** carbon fibre reinforced polymer (CFRP), thermoplastic composites, high performance composites, impact damage, non-crimp fabric (NCF), finite element analysis (FEA)

## Abstract

There has been a lot of interest in understanding the low-velocity impact (LVI) response of thermoplastic composites. However, little research has focussed on studying the impact behaviour of non-crimp fabric (NCF)-based fibre reinforced thermoplastic composites. The purpose of this study was to evaluate the LVI responses of two types of non-crimp fabric (NCF) carbon fibre reinforced thermoplastic laminated composites that have been considered attractive in the automotive and aerospace industry: (i) T700/polyamide 6.6 (PA6.6) and (ii) T700/polyphenylene sulphide (PPS). Each carbon/thermoplastic type was impacted at three different energy levels (40, 100 and 160 J), which were determined to achieve three degrees of penetrability, i.e., no penetration, partial penetration and full penetration, respectively. Two distinct non-destructive evaluation (NDE) techniques ((i) ultrasonic C-scanning and (ii) X-ray tomography) were used to assess the extent of damage after impact. The laminated composite plates were subjected to an out-of-plane, localised impact using an INSTRON^®^ drop-weight tower with a hemispherical impactor measuring 16 mm in diameter. The time histories of force, deflection and velocity are reported and discussed. A nonlinear finite element model of the LVI phenomenon was developed using a finite element (FE) solver LS-DYNA^®^ and validated against the experimental observations. The extent of damage observed and level of impact energy absorption calculated on both the experiment and FE analysis are compared and discussed.

## 1. Introduction

Laminated composite structures used for aerospace and automotive applications have always been susceptible to damage and failure due to in-plane loading conditions such as tension and compression and out-of-plane contact such as impact with foreign objects [1]. For example, the exterior components of a vehicle, such as the bumper, fender and bonnet, are constantly prone to impact, particularly low-velocity transverse impact. Impact damages as such could result in matrix cracking, fibre fracture and delamination [2,3,4], all of which lead to a deterioration of the mechanical properties of the material. Such damage could also be very difficult identify with the naked eye.

Recently, the automotive industry has shown a significant interest in understanding the mechanical behaviour of thermoplastic composites (TP) [5,6] due to their out-of-autoclave (OOA) manufacturability and recyclability, which are not currently achievable with typical thermosetting (TS) systems. In addition, since thermoplastic-based composites are more readily recyclable [7], thermoplastics, including thermoplastic composites, conform with EU directive 2000/53/EC [8], which has a target whereby at least 85% of the total materials used on the average vehicle per year should be reused or recycled.

Hence, there is a need to characterise thermoplastic (TP) composites’ response under impact to allow for better prediction of their behaviour under different structural applications, e.g., for automotive and aerospace applications. This research aims to contribute invaluable information to the ever-growing composite materials database.

### 1.1. Characterisation of Impact Behaviour of Thermoplastic Composites

Carbon fibre reinforced polymer (CFRP), despite the attractive in-plane properties that it offers, are sensitive and considerably weaker under out-of-plane loads. Delamination has been considered one of the largest factors [9].

Currently, the available data and information in the literature with respect to the low-velocity impact (LVI) performance of carbon fibre reinforced thermoplastic polymer (CFRTP) is still limited, unlike carbon fibre reinforced thermosetting (CFRTS) composites [1,10,11,12,13,14,15,16,17,18]. In fact, in the literature, much research has focused solely on glass fibre reinforced polymer (GFRP) [19].

Trellu et al. [10] have previously reported the compression after impact (CAI) performance of unidirectional (UD) CFRTS (T700/M21 epoxy) at velocities ranging from 54 m/s to 110 m/s. This represents a small debris impact during aircraft take-off or landing. Under those loading conditions, it was claimed that the failure was dominated by shear loading. A comparison between the impact performance between woven CFRTP and CFRTS has also been reported [20] (under 2 J to 25 J impact energy). Vieille et al. [20] concluded that the tougher matrix in the CFRTP can be associated with the superior impact performance of CFRTP over CFRTS.

Hitchen and Kemp [13], on the other hand, have discussed the effect of stacking sequences on the impact damage (with 7 J impact energy using a 2 kg mass, velocity of 2.5 m/s) of CFRTS. It was found that the stacking sequence has a major influence on the total delamination area and the energy absorbed during impact. Baba et al. [14] also presented an experimental study on the LVI of 2.5 mm thick, 150 mm × 100 mm laminated CFRTS panels. The LVI experiment was carried out at velocities ranging from 1 m/s to 5 m/s, which corresponds to 23 J impact energy (using a 1.9 kg impactor). However, their analytical approach (using classical laminate theory) was only able to produce an accurate prediction for impact velocities of 1 m/s. Clearly, FE modelling would allow for enhanced prediction of the composite behaviour.

LVI testing of CFRTP has been reported, but the information in the open literature is still scarce. For example, LVI testing of carbon fibre/polyamide 6 (PA6) and carbon fibre/polyamide 66 (PA6.6) have been reported by [21,22]. Bondy and Altenhof [22] carried out an LVI test on 600 mm × 600 mm CFRTP (carbon/PA6.6) panels using a 60 kg carriage and a 20 mm impactor with a mean impact velocity of 4.4 m/s, which resulted in an impact energy of 570 J. Strain-rate sensitivity was also reported on the carbon/PA6.6 material system [22]—550% higher stiffness was observed in LVI versus quasi-static loading. This supported the claim made by Mohsin et al. [23] for virtually identical CFRTP system. However, the studies reported in [21,22] are not directly applicable to the automotive or aerospace industry as the impactor mass is too high and the impact velocity is too low (despite the high impact energy).

Another interesting study on LVI behaviour of CFRTP has been performed using carbon/polyether ether ketone (PEEK) [24]. Here, Vieille et al. reported the impact behaviour of hybrid carbon and glass fibre reinforced PEEK using Charpy pendulum and drop tower tests. This was carried out using 4.5 mm thick 150 mm × 150 mm composite laminates with 2 kg, 16 mm impactor at velocities ranging from 2 m/s to 6 m/s, resulting in the impact energies ranging from 25 J to 40 J. There has also been a study on LVI of NCF CFTRP with novel liquid Methylmethacrylate (MMA) TP matrix, in which the laminates were impacted at 25 J, 42 J and 52 J [25]. The results were then compared to a baseline CFRTS (carbon/epoxy) and indicated higher energy absorption in the CFTRP.

Thus far, the range of velocities and impact energies is limited. Hence, there is a gap in the literature to investigate a wider range of impact velocities (e.g., 5 m/s to 10 m/s) and impact energies (e.g., 40 J to 160 J). In addition, there is also a need to study the LVI behaviour of NCF biaxial CFRTP, as most of the previous research on LVI revolved around UD, multilayer UD and woven fibre architecture. To the best of the author’s knowledge, no information on the LVI of CFRTP carbon/PPS was found.

Therefore, due to the ongoing interest from different industries, particularly automotive and aerospace (as well as the industrial partners of Imperial College), the experimental and numerical analysis (methodology) presented in this research will contribute towards the further understanding of CFRTPs. Based on the literature review, it was clear that there was a lack in the not just the experimental data of LVI behaviour of CFRTP, but also the accurate numerical model (and modelling technique) that thoroughly describe the phenomenon.

### 1.2. Numerical Model Development for Impact Behaviour Prediction

To date, the available literature on the numerical model development for the prediction of impact performance of non-crimp fabric (NCF) CFRTP are very limited. Patil and Mallikarjuna Reddy [11] have presented the oblique impact loading of another CFRTS at impact velocities between 3 m/s and 10 m/s and an explicit FE model using Abaqus (Dassault Systèmes, Vélizy-Villacoublay, France) was successfully developed to describe the LVI behaviour. This was proven to be beneficial in predicting (within limits) the real-world impact performance of the material system.

Hou et al. [26] proposed a novel multiscale modelling strategy of the LVI behaviour of plain woven composites using an equivalent cross-ply laminate. However, this was performed on and validated for a CFRTS carbon/epoxy system. Additionally, it was only validated against two initial velocities: 2.83 m/s and 3.46 m/s, which corresponded to impact energies of 4 J and 6 J, respectively. Nevertheless, the novel numerical model implemented provided good correlation with the experimental results.

Liu [27] also presented a damage model and numerical method for predicting the dynamic progressive failure of composite laminates under LVI loading conditions, specifically for two CFRTS systems: (i) T300/M21 carbon/epoxy and (ii) HS300/ET223 graphite/epoxy. The approach was performed using Abaqus-Python scripting language and a numerical technique that was developed by Abaqus-VUMAT explicit material module. It was reported that the Puck criteria-based model yielded minimal discrepancy with experiments when compared to the more typical approaches used in commercial FE solvers, such as Abaqus and LS-DYNA^®^. Commercial solvers are known to adopt the simpler implementation using Hashin and Chang-Chang criteria but offered a good balance between numerical precision and computational cost. Hence, there is a need to further investigate the current capabilities of commercial FE solvers in particular.

## 2. Materials and Manufacturing Methods

### 2.1. Material System and Preparation

Two CFRTP material systems were used in this research: (i) NCF biaxial (0°/90°) T700 (continuous) carbon pre-impregnated with polyamide 6.6 veils (T700/PA6.6) and PA6.6 stitching; and (ii) NCF biaxial (0°/90°) T700 (continuous) carbon pre-impregnated with polyphenylene sulphide veils (T700/PPS) and Kevlar^®^ (DuPont, Wilmington, DE, USA) stitching. The raw materials were supplied by THERMOCOMP project [6] partners. The T700/PA6.6 material system has also previously been reported and discussed in Mohsin et al. [23].

The material (unreinforced and reinforced) and mechanical properties of the constituent materials of the composite are shown in Table 1. However, since the material system is proprietary, the mechanical properties of the laminates were obtained by the author using a series of standardised and non-standardised tests listed in Table 2 and described in [28].

To eliminate the effects of moisture, the densities of both CFRTP material systems were measured using a pycnometer after being stored in an oven at 40 °C for three days. The densities of the T700/PA6.6 and T700/PPS systems were measured to be 1485 kg/m^3^ and 1553 kg/m^3^, respectively.

The fibre-volume-fraction (FVF) of the laminates produced was measured via thermogravimetric analysis (TGA) (T700/PA6.6 = 52% and T700/PPS = 61%). The process has been detailed in [34].

### 2.2. Material Preparation, Manufacturing Process and Specimen

The CFRTP laminates were prepared using a hand lay-up method and manufactured using a thermoforming method using a laboratory hydraulic (HÖFER, Taiskirchen, Austria) press at 275 °C. Each laminate (rectangular panel) comprised 24 plies of the T700/PA6.6 with the following layup sequence: (0/90)_12s_.

For the T700/PA6.6, the manufacturer’s recommended processing parameters are as follows: Dwell time: 10 minProcessing temperature: 275 °CHeating rate: 15 °C/minPressure: 1.5 MPaDemoulding temperature: 25–35 °C.

The average thickness of the T700/PA6.6 panel was measured to be 4.1 ± 0.28 mm.

For the T700/PPS, the manufacturer’s recommended processing parameters are as follows: Dwell time: 10 minProcessing temperature: 315 °CHeating rate: 15 °C/minPressure: 2.5 MPaDemoulding temperature: <100 °C.

The average thickness of the T700/PPS panel was measured to be 4.3 ± 0.34 mm. Despite the varying demoulding temperature recommended by the manufacturer, the demoulding procedure has always been carried out at ambient temperature, which was typically around 25 °C.

## 3. Low-Velocity Impact Test and Experimental Setup

### 3.1. Low Velocity Impact Test

The LVI test allows for the determination of the damage resistance of a laminated composite subjected to a drop-weight impact event. The test was conducted partially in accordance with the standardised test, ASTM D7136/D7136M [16]. The dimensions of the drop-weight impact test panel and impact location were as described in Figure 1. The test method was designed to characterise materials for damage resistance and tolerance. The impact performance of a laminated composite material is largely governed by several common factors, such as specimen geometry, layup quality, impactor mass, force and energy, and boundary conditions. Hence, it must be noted that the results gathered specifically from this test are not necessarily scalable to other configurations.

At present, the published information with regard to the LVI performance of carbon fibre reinforced thermoplastic (CFRTP) is scarce, unlike TS composites [1,12,13,35]. Additionally, since the automotive industry has been continually interested in finding alternatives to OOA manufacturing, this study aims to provide invaluable information with respect to the impact resistance of TP composites and how they compare to their TS counterparts.

### 3.2. Experimental Setup

The drop-weight impact test was performed using a balanced, symmetrical laminated composite plate. The damage was induced out-of-plane, aligned on the centre of the plate using a hemispherical impactor with a diameter of 16 mm from an INSTRON^®^ drop tower machine (Instron Corporation, MA, USA) (Figure 2). The impact response or damage resistance was measured in terms of the damage, type and size of the panel.

Three energy levels were chosen (40, 100 and 160 J) to achieve three different degrees of penetrability: no penetration, partial penetration and full penetration. The energies were chosen based on initial (rough) calculations and past experience of working with 4 mm-thick panels. The impact velocity, impactor displacement and applied contact force against time history were recorded.

The impact energy absorption was calculated by partially integrating the area under the force-displacement graph. Nonetheless, the energy absorption was only calculated for when the damage started to occur, which refers to only two energy levels, 100 and 160 J. The calculated values reported in Table 1 specifically represent the initiation impact energy. Hence, the area under the curve of interest consists of the beginning of impact, where the force starts to increase to the point where it starts to decrease.

## 4. Experimental Results and Discussion

### 4.1. Experimental Observations: Extent of Damage, Ultrasonic C-scan and X-ray Images of the Specimens Post Impact 

The laminated composite panels post impact are shown in Figure 3. This figure highlights the different types and degrees of damage, which include typical damage characteristics, such as delamination, matrix cracking and fibre breakage. 

Following the test, each panel was inspected using ultrasonic C-scan and X-ray tomography. The images obtained from the ultrasonic C-scan is shown in Figure 4. The NDE scans of the representative panels were also complemented by the X-ray images shown in Figure 5 to further visualise the extent of the damage to the materials.

The NDE of each specimen representing the varying impact loading conditions exhibited relatively predictable levels of damage, where the lowest to the greatest correspond to the lowest to the highest impact energy, i.e., 40 to 160 J. Likewise, the X-ray images revealed similar results albeit with better clarity. Using an open-source Java-based image processing program, ImageJ (National Institutes of Health, Bethesda, MD, USA), the outline of the X-ray images was marked (Figure 6) and the damage area was calculated for each laminate.

The C-scan (Figure 4) and X-ray images (Figure 5) illustrate that the size of damage increases as the energy level increases. At 40 J, all the specimens indicate a minor extent of damage. With regard to the T700/PA6.6 samples, the damage was found to be most localised as the extent of damage was the smallest, particularly at 100 and 160 J. Additionally, significant petalling of the rear surface was observed. This was analogous to the reports found in the literature [20,22]. Conversely, Figure 5 and Figure 6 show that the T700/PPS specimens merely suffered severe delamination but no penetration.

### 4.2. Experimental Results

Figure 7, Figure 8 and Figure 9 depict the time histories (force–time, deformation–time and velocity–time) and a force–displacement graph of the T700/PA6.6 and T700/PPS CFRTP material systems under varying impact conditions. These were obtained from the data acquisition system that was connected to the drop tower.

Figure 7 shows that under an impact energy of 40 J (i.e., no penetration), the material seemed to behave comparably. However, at higher impact energies, e.g., 100 J (Figure 8) and 160 J, each of the two materials tested indicated distinct behaviours. 

At 100 J, the representative T700/PA6.6 specimen appears to experience the highest magnitude of impact force, followed by the T700/PPS samples (Figure 8). With respect to deformation, the T700/PPS specimen seems to suffer greater deformation than the T700/PA6.6.

With regard to the time histories and force–deformation curve at 160 J (Figure 9), once again, the T700/PA6.6 panel indicates the largest impact force, while the T700/PPS experienced a comparable magnitude of impact force. In relation to the degree of deformation, the force–displacement curve obtained in Figure 9 suggests that the T700/PA6.6 sample shows the greater level of deformation than T700/PPS specimen.

Since the thicknesses of the impact panels vary slightly (with a coefficient of variation, CV = 7%) across the different material systems, the calculated values of energy absorption, i.e., the conforming area under the force-displacement curve, were normalised to the respective areal weight of each system. 

PA6.6 (the more commonly used matrix in the automotive industry) is universally known to be the weaker thermoplastic polymer. PPS, on the other hand, is typically recognised as an aerospace-grade matrix (that is more expensive). Therefore, it was useful to compare the two sets of CFRTPs, setting the T700/PA6.6 as the baseline and see how much more impact performance could be obtained with the T700/PPS. To do this, the impact energy absorbed by each panel must be normalised against the areal weight of the material system. This is illustrated in Table 3, where the LVI performance of the two CFRTP systems were compared with respect to their accompanying densities and areal weights. 

### 4.3. Discussion

The impact damage seen in the laminates exhibits a combination of matrix cracking, fibre matrix debonding, delamination and fibre fracture. Delamination, for instance, occurs due to the low interlaminar shear strength, which leads to a significant reduction in the material’s performance after impact. Laminated composite structures are habitually designed to absorb such LVI in most structural applications. When this laminated structure is exposed to barely visible impact damage (BVID), micro-damage is sustained. This leads to a critical reduction in the laminate’s strength and durability [13,38,39,40].

At the lowest impact energy (40 J), all systems showed comparable responses, as expected (Figure 3). This is because at 40 J, only BVID was observed by the naked eye. However, the C-scans and X-ray images show otherwise (Figure 4, Figure 5 and Figure 6), where the damage on 40 J panels is evident. Similarly, force-displacement (Figure 7) plots did not return to the origin, indicating a plastic failure within the laminate.

Under the intermediate impact energy of 100 J, it was initially predicted that the samples would all suffer partial penetration. However, this was only achieved on the T700/PA6.6 panel with the petalling effect on the rear surface, not on the T700/PPS. On the contrary, the T700/PPS system showed a large degree of delamination, but without any penetration. The extent of damage in the T700/PA6.6 sample at 100 J indicated by the C-scan images appear to be smaller and more localised, with smaller delamination in comparison to the T700/PPS. The extent of damage and delamination in the T700/PPS sample were found to be more severe. Nevertheless, the T700/PPS system was able to absorb more (+14.8%) energy per areal weight compared to the T700/PA6.6. This was obtained from the calculated area under the force-displacement plot.

At 160 J, as expected, the extent of damage and delamination seen at 100 J were simply amplified. The T700/PA6.6 laminates underwent full penetration whereas the T700/PPS suffered a greater degree of delamination. This was clearly shown in both the C-scans and X-ray images. Overall, the T700/PPS still showed 13.3% more energy absorption when compared to the T700/PA6.6 system.

The delamination areas of T700/PPS relative to the T700/PA6.6 are quantified in Table 4. The relative damage areas of the T700/PPS (with respect to T700/PA6.6) were calculated to be at 3.2, 1.8 and 1.2 at 40 J, 100 J and 160 J, respectively.

Since delamination of a laminated composite is largely governed by the interlaminar shear stresses, it is one of the main energy absorption features of polymer composite materials. Hence, the results obtained from this study are indicative of the interlaminar shear strength reported by Mohsin et al. [36], where the weakest T700/PPS suffered the greatest level of delamination while absorbing the highest amount of energy per areal weight. It is also postulated that the delamination that was seen in the T700/PPS system under both impact energies of 100 J and 160 J could have been contributed by the relatively tougher Kevlar^®^ stitching in the NCF. The stitching may have also resulted in a *pull-in* effect seen on the post-impact T700/PPS panels. The T700/PA6.6 exhibited a more localised penetration with a comparatively small extent of damage and low degree of delamination. This is indicative of its higher interlaminar shear strength when compared to the T700/PPS.

## 5. Finite Element Model of the Low-Velocity Impact Test

### 5.1. Model Description

Commercial finite element (FE) solver LS-DYNA^®^ R8.1.0 (R8.105896) (LSTC, Livermore, CA, USA) was used to perform the finite element analysis (FEA) throughout this research. The FE model was developed and aimed to simulate the LVI test. 

To compare, calibrate and validate the FEA with experimental results, force–time, velocity–time and displacement–time histories were exported. Each panel’s damage response was observed and compared against the experimental gathering.

#### 5.1.1. Impactor and Rig

The FE model comprised a hemispherical impactor and a composite panel with boundary conditions that satisfy the actual test. The impactor’s mass was kept constant at 2.456 kg. The impact energies were set by changing the impact velocity.

In the experiment, a hemispherical steel impactor with a diameter of 16 mm was used to impact the laminate. The impactor was mounted onto the carriage on the drop tower. In the FE simulation, however, the steel impactor was shortened and its density was increased to compensate for its lack of size and to achieve the actual mass of the impactor. The impactor was modelled as a rigid body (i.e., RIGID elements). The boundary conditions of the impact rig in the FE was set to exactly match the conditions set in the actual experiment, where it was laterally constrained.

#### 5.1.2. Composite Panel

The model of the panel was prepared using the continuum shell (TSHELL) elements to represent both the T700/PA6.6 and T700/PPS laminates. This was constructed using four layers of continuum shell elements with six integration points in each layer to represent the total of 24 plies of the T700/PA6.6 and T700/PPS, as shown in Figure 10. The material card used to predict the composite behaviour was the energy-based MAT_262-LAMINATED_FRACTURE_DAIM-LER_CAMANHO. 

Prior to determining the optimised mesh size (of 1 mm), a mesh dependency study was carried out using element sizes ranging from 0.5 to 2.0 mm (Figure 11). The mesh sensitivity study was carried out for a panel subjected to the impact energy of 40 J. The results obtained from the FE were then compared to the experimental gathering and the effects of the mesh size on the load-time histories (Figure 12) and the load–displacement relationship (Figure 13) with respect to the maximum load and the profile of the plot. 

Based on this, it was concluded that the element size of 1 mm was optimal for the specific setup generated in the FE. The summary of the element size, number of elements used, computational time and maximum load (observed) is listed in Table 5. The job was carried out using a quad core, hyper-threaded Intel^®^ Core i7-4930 MX (Intel^®^, Santa Clara, CA, USA).

The modelling parameters used to represent both material systems are show in Table 6. These properties have been decoupled to meet the requirements of the selected material card.

#### 5.1.3. Contact

The contact algorithm, i.e., the cohesive surface used to simulate the thermoplastic veils in the composite panel, was the AUTOMATIC_SURFACE_TO_SURFACE_TIEBREAK. It has been found that a higher number of cohesive interfaces reduces the plate bending stiffness, which can produce substantially different impact responses, specifically at the beginning of impact [40]. Therefore, it was determined empirically that three cohesive surfaces should be used between the four layers of continuum shell elements. Cohesive elements could also be used instead, but this method was found to be computationally more expensive. The properties of the (cohesive) contact surface are listed in Table 7.

The damage initiation in the cohesive surface is predicted using the discrete crack model with power law and B-K damage models. As shown in Equation (1) [41], the mixed-mode damage initiation displacement, $\delta^{0}$ is calculated as:(1)$$\delta^{0}=\delta_{I}^{0}\delta_{II}^{0}\sqrt{\frac{1+\beta^{2}}{{\left(\delta_{I}^{0}\right)}^{2}+{\left(\delta_{II}^{0}\right)}^{2}}}$$
where $\delta_{I}^{0}=T/EN$ and $\delta_{II}^{0}=S/ET$ are the single mode damage initiation separation and $\beta =\delta_{II}/\delta_{I}$ is the ‘mode mixity’, based on the mixed-mode traction-separation law [41].

Therefore, the ultimate mixed-mode displacement $\delta^{F}$ (total failure) for the power law (XMU > 0) is as follows:(2)$$\delta^{F}=\frac{2{\left(1+\beta \right)}^{2}}{\delta^{0}}{\left[{\left(\frac{EN}{G_{IC}}\right)}^{XMU}+{\left(\frac{ET.\beta^{2}}{G_{IC}}\right)}^{XMU}\right]}^{-\frac{1}{XMU}}\;$$

The default exponent damage in LS-DYNA^®^, XMU = 1, where the damage is assumed to be purely in mode I. However, in LVI the mode II damage is typically dominant. Hence, following the calibration process, the best value (for XMU) in this specific scenario was found to be 1.8—this was calculated using Equation (2) [41].

#### 5.1.4. Boundary Conditions and Prescribed Motion

The boundary conditions were set to simulate the actual experiment. Therefore, the rig was fixed in all translational (X = 0, Y = 0, Z = 0) and rotational directions (RX = 0, RY = 0, RZ = 0). The impactor was set to be able to move freely in the Z-direction but fixed everywhere else to accurately simulate the experimental setup (i.e., X = 0, Y = 0, RX = 0, RY = 0, RZ = 0). The impactor was prescribed varying initial velocities to achieve the desired impact energy level (Table 8).

### 5.2. Results and Discussion

Figure 14 and Figure 15 illustrate the load–time histories and load–displacement curve for T700/PA6.6 and T700/PPS, respectively. In these figures, FE results are indicated in red and experimental data are shown in varying shades of grey. At 40 J, it is evident that the FEA and experimental gathering showed excellent correlation. The amount of energy dissipated by the impact in the LVI when compared to the FE simulations is very comparable.

At 100 J, the correlation between the FE and experimental results is still reasonably similar since the FE results lie within the variation observed in the experiments (indicated by the three lines in different shades of grey). The load–time histories and load–displacement curves for both material systems only started to deviate upon damage propagation. This disparity seen in the T700/PA6.6 was due to the shear-plugging phenomenon observed during the experiment, which was clearly not accurately captured in the FE and could not be accounted for using continuum shell elements. In relation to the T700/PPS, the discrepancy was instigated by the severe delamination seen in the experiment, which also could not be thoroughly accounted for by the FE model.

At 160 J, the trend in both load–time histories and load–displacement curves was found to deviate further from the experiment in comparison to the results at 100 J. This is expected as the higher the LVI energy, the greater the effect of shear-plugging and delamination seen in the T700/PA6.6 and T700/PPS. Once again, this is primarily caused by the limitations of the model, specifically the continuum shell elements. The continuum shell elements failed to reproduce realistic through-thickness behaviour of the actual experiment, hence the disparity during the onset of damage propagation.

Figure 16 and Figure 17 represent the damage areas observed in the FEA and LVI experiments (obtained from X-ray) of the T700/PA6.6 and T700/PPS, respectively. Naturally, the damage areas observed from FE results appear to be more regular and symmetrical (especially with respect to the T700/PA6.6 laminates). This is because the FE model assumes that the material properties of both materials are uniformly orthotropic throughout the laminate, which is not the case in reality due to manufacturing quality and defects.

Based on Table 9, it can be observed that the disagreements between the FE and experimental results of T700/PA6.6 were found to be about 15%. However, at 160 J, the difference in the damage area is larger, at 35%, which is expected at the higher energy level, where the extent and nonlinearity of damage are greater. At 160 J, the panel suffered a more complex failure mode, dominated by the shear-plugging in the T700/PA6.6.

With respect to the T700/PPS, it was discovered that the discrepancies between the FE and experimental results are greater than that of the T700/PA6.6. The primary reason for this is the inferior interlaminar shear properties of the T700/PPS. Shear properties of laminated composites play one of the most important roles in determining the material’s LVI response. The weaker interlaminar shear properties of the T700/PPS make the material more susceptible to delamination failure. In the FE model, the through-thickness properties at the interface inhibit the delamination initiation and propagation. The use of continuum shell exacerbates this as the through-thickness behaviour of the material is not properly accounted for. However, this can be resolved using solid elements, albeit at a higher computational cost with more complex failure and contact formulation.

Nevertheless, the numerical model’s accuracy can also be quantified by comparing the peak loads obtained from FE against the experimental results. In this way, a reasonable comparative analysis can be carried out to measure the accuracy of the numerical model. With reference to Figure 18, it can be concluded that the peak loads predicted by the FE at the different LVI energies lie within the variation seen in the experiments. Thus, the FE model was able to forecast the load required to initiate damage very accurately.

## 6. Conclusions

The LVI performance of both CFRTP systems have been compared at three different energy levels (40, 100 and 160 J) experimentally and numerically. This was done to achieve no penetration, partial penetration and full penetration. At the lowest energy level (40 J), BVID has been identified using two NDE techniques: ultrasonic C-scanning and X-ray tomography. This is consistent with the information discovered in the literature [18], where laminated composites are sometimes prone to BVID when incurring micro-damage.

At the intermediate energy level (100 J), a shear-plugging phenomenon was observed in the T700/PA6.6 and severe delamination was seen in the T700/PPS. This is in line with the findings in the literature where interlaminar properties govern the delamination of laminated composites during LVI. The inferior interlaminar properties of the T700/PPS compared to the T700/PA6.6 resulted in significant delamination in the material system. The T700/PPS absorbed about 15% more energy per areal weight at 100 J. 

At LVI energy of 160 J, the extent of damage seen in both systems was more pronounced. The T700/PA6.6 continued to exhibit localised penetration with a comparatively small extent of damage and a low level of delamination and the T700/PPS suffered a higher degree of delamination with no penetration. As a result, the T700/PPS showed a 13% improvement over the T700/PA6.6.

The LVI FE model was developed using an energy-based fracture model in a commercial solver LS-DYNA^®^. The numerical model exhibited excellent correlation with the experimental results of the impact response at the lowest impact energy level of 40 J. At 100 J and 160 J, good agreement with experimental gatherings was found when forecasting the peak load, i.e., onset of damage initiation. However, there were discrepancies when comparing the damage propagation in FEA and experimental data. Due to the limitations of the FE model, specifically its inability to fully capture the through-thickness behaviour of the laminate using continuum shell elements, larger discrepancies were found at higher energy levels. The model was unable to predict complex responses such as shear-plugging and extreme delamination. 

As explained in this paper, the challenges involved in developing a more accurate model of LVI behaviour is twofold; the computational cost of using solid elements and the need for a user-defined formulation that can account for the complex failure.

## Figures and Tables

**Figure 1 polymers-13-03642-f001:**
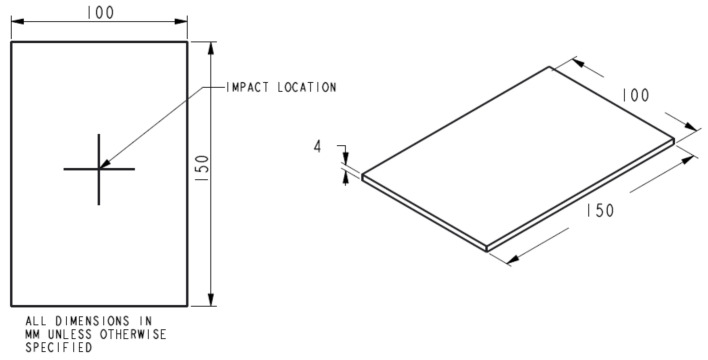
Drop-weight impact test specimen according to ASTM D7136/D7136M [37].

**Figure 2 polymers-13-03642-f002:**
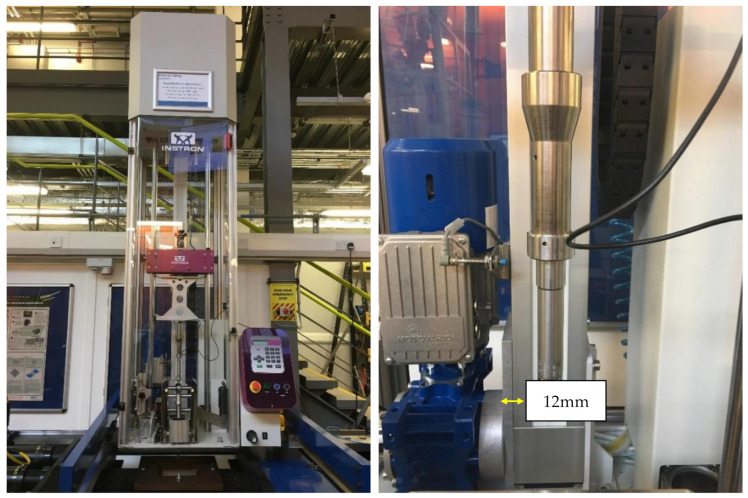
INSTRON^®^ drop tower (**left**) and close-up image of the impactor (**right**).

**Figure 3 polymers-13-03642-f003:**
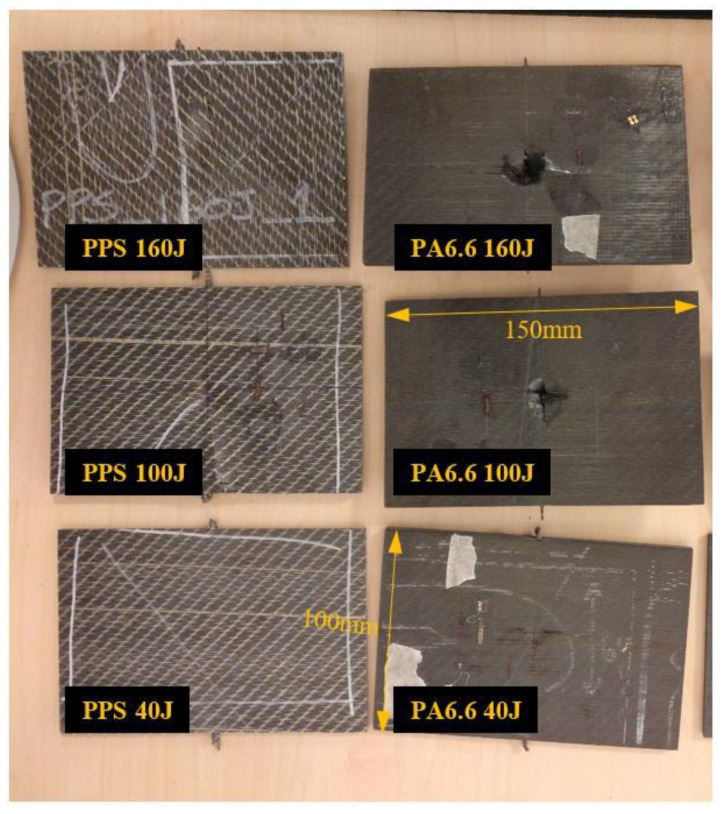
The laminated T700/PPS (**left**) and T700/PA6.6 (**right**) specimens post impact.

**Figure 4 polymers-13-03642-f004:**
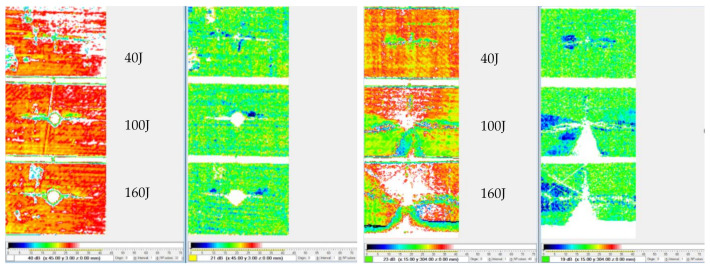
C-scans of the T700/PA6.6 (**left**) and T700/PPS (**right**).

**Figure 5 polymers-13-03642-f005:**
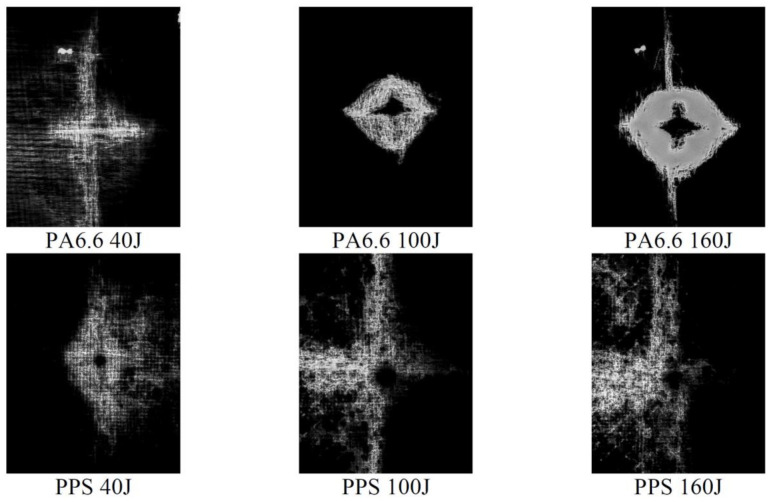
X-ray images of the specimens after impact.

**Figure 6 polymers-13-03642-f006:**
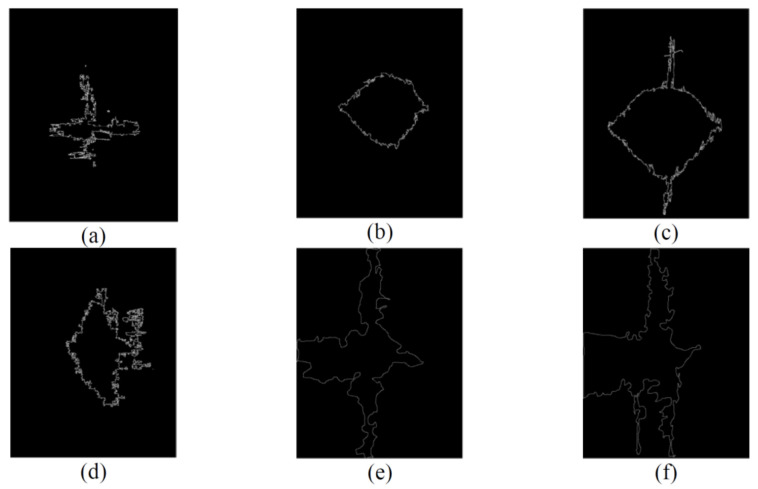
Damage area on the T700/PA6.6 post-LVI at 40 J, 100 J and 160 J: (**a**) 683 mm^2^, (**b**) 1502 mm^2^ and (**c**) 2705 mm^2^; T700/PPS post-LVI at 40 J, 100 J and 160 J: (**d**) 1781 mm^2^, (**e**) 6686 mm^2^ and (**f**) 7098 mm^2^.

**Figure 7 polymers-13-03642-f007:**
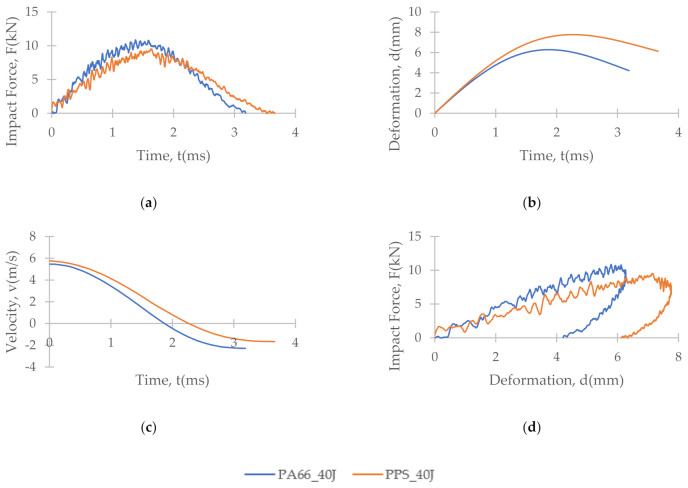
Time histories: (**a**) force-time, (**b**) deformation-time and (**c**) velocity-time; and (**d**) force-displacement plot of T700/PA6.6 and T700/PPS CFRTP composite systems under 40 J impact energy.

**Figure 8 polymers-13-03642-f008:**
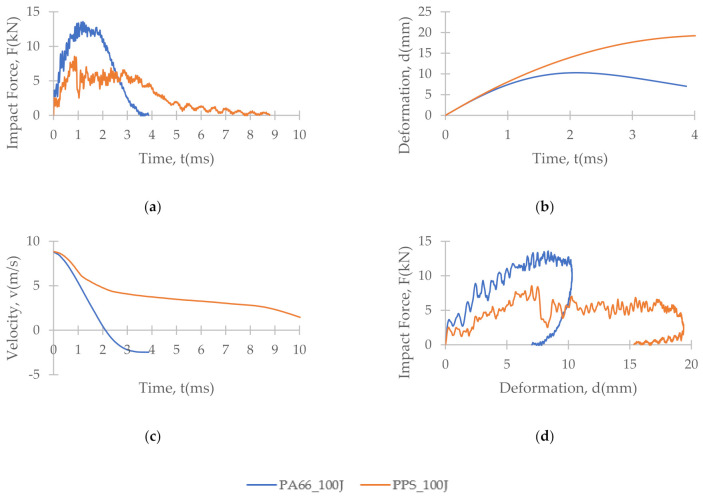
Time histories: (**a**) force-time, (**b**) deformation-time and (**c**) velocity-time; and (**d**) force-displacement plot of T700/PA6.6 and T700/PPS CFRTP composite systems under 100 J impact energy.

**Figure 9 polymers-13-03642-f009:**
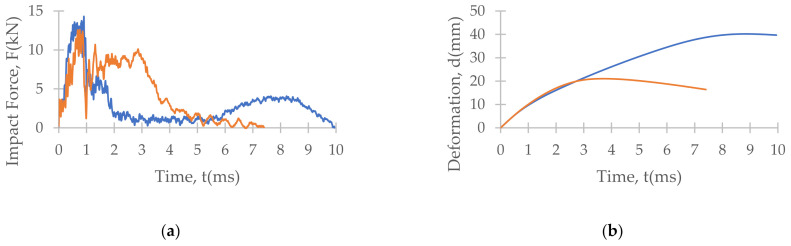
Time histories: (**a**) force-time, (**b**) deformation-time and (**c**) velocity-time; and (**d**) force-displacement plot of T700/PA6.6 and T700/PPS CFRTP composite systems under 160 J impact energy.

**Figure 10 polymers-13-03642-f010:**
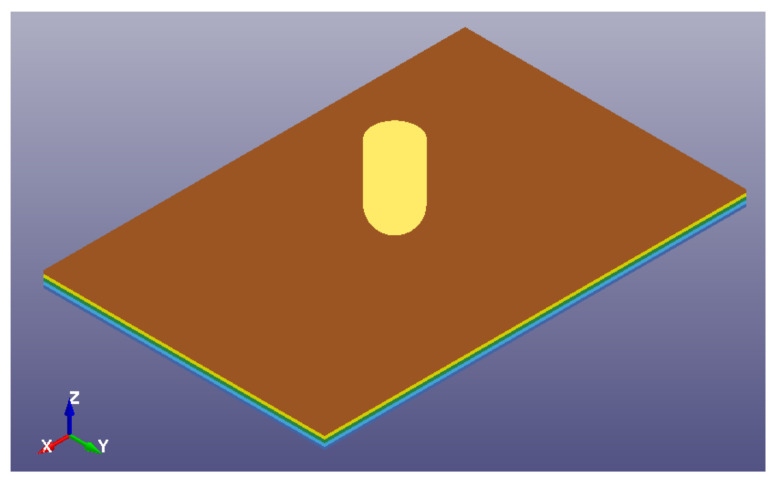
FE model of the LVI setup in LS-DYNA^®^ [28].

**Figure 11 polymers-13-03642-f011:**
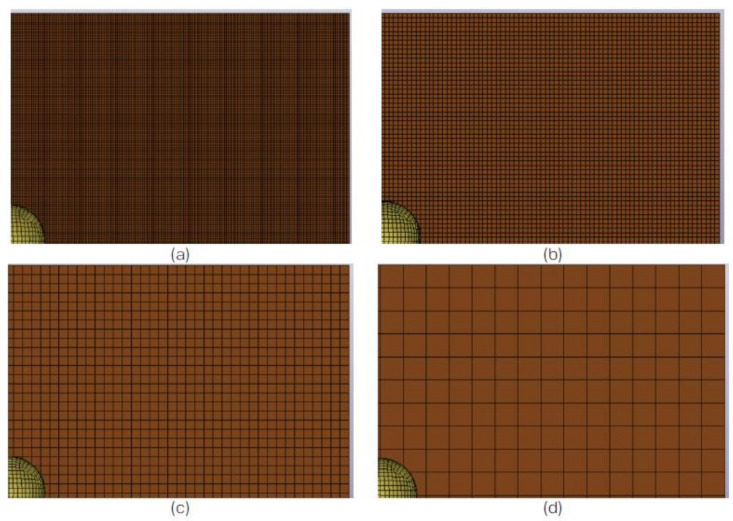
View of a quarter of the FE model for the LVI panel created with different mesh/element size: (**a**) 0.5 mm, (**b**) 1.0 mm, (**c**) 2.0 mm and (**d**) 5.0 mm [28].

**Figure 12 polymers-13-03642-f012:**
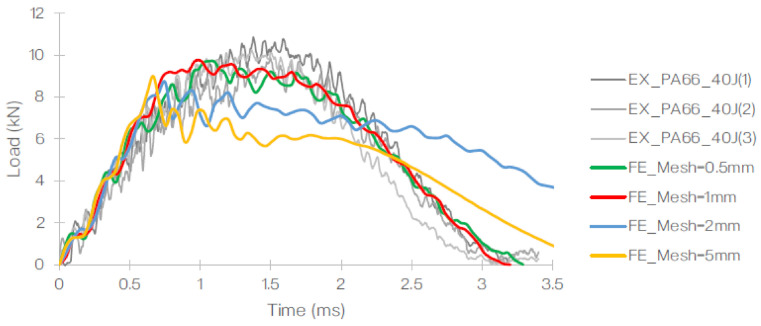
Comparison of the load–time curve obtained from the FE for the 40 J impact event conducted using different mesh sizes against the experiment [28].

**Figure 13 polymers-13-03642-f013:**
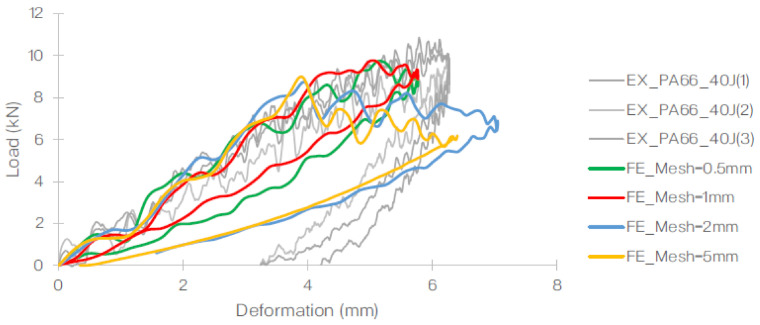
Comparison of the load–displacement curve obtained from the FE for the 40 J impact event conducted using different mesh sizes against the experiment [28].

**Figure 14 polymers-13-03642-f014:**
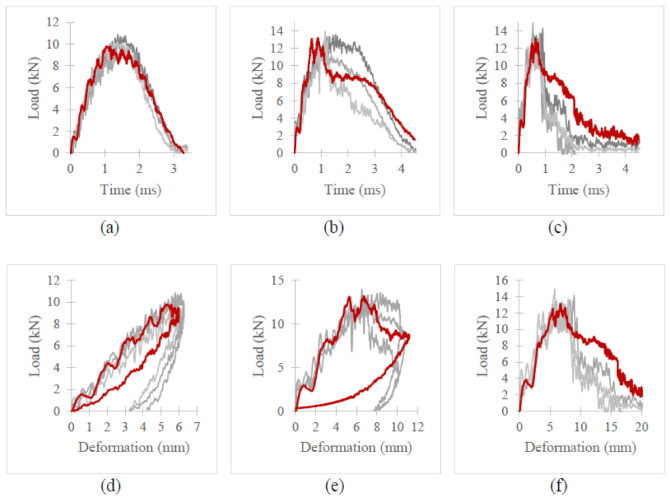
Force–time histories (**a**–**c**) and force–displacement curves (**d**–**f**) for T700/PA6.6 at 40, 100 and 160 J.

**Figure 15 polymers-13-03642-f015:**
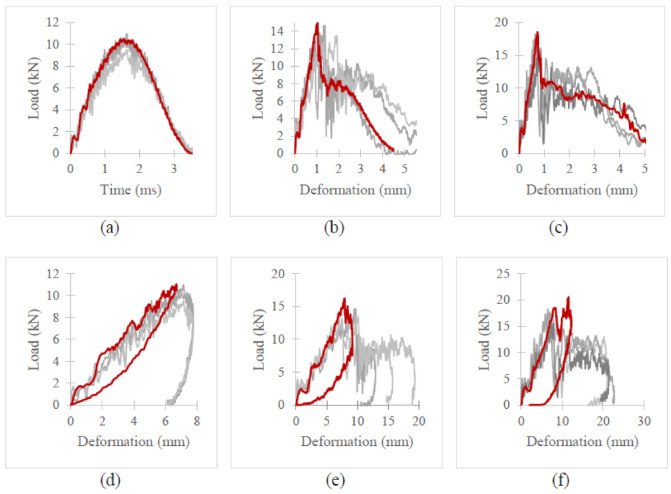
Force–time histories (**a**–**c**) and force–displacement curve (**d**–**f**) for T700/PPS at 40, 100 and 160 J.

**Figure 16 polymers-13-03642-f016:**
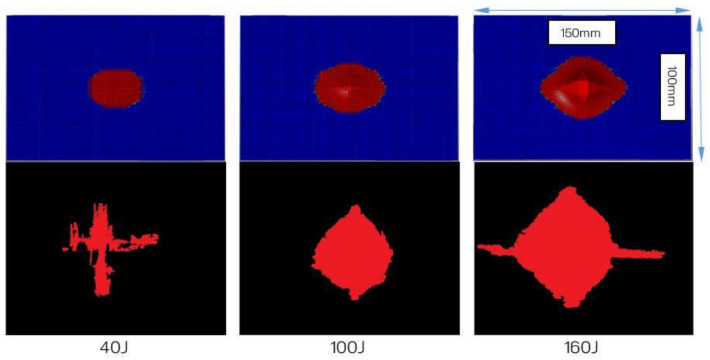
Damage area (envelope) computed by the numerical simulations (top three images) and damage area gathered following the experiments through X-rays for T700/PA6.6. Each image represents an overall panel with dimensions of 150 mm × 100 mm.

**Figure 17 polymers-13-03642-f017:**
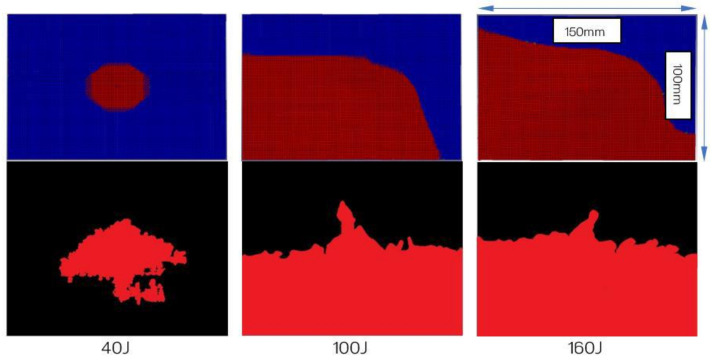
Damage area (envelope) computed by the numerical simulations (top three images) and damage area gathered following the experiments through X-rays for T700/PPS. Each image represents an overall panel with dimensions of 150 mm × 100 mm.

**Figure 18 polymers-13-03642-f018:**
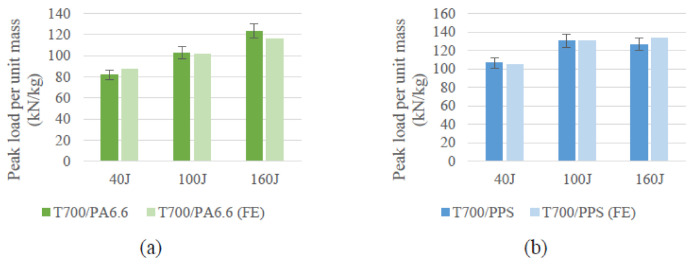
Comparison between experiments and FE with respect to peak load per unit mass for the (**a**) T700/PA6.6 and (**b**) T700/PPS.

**Table 1 polymers-13-03642-t001:** Mechanical properties of neat PA6.6, PPS and T700 fibre.

	PA6.6	PPS	T700 Fibre
**References**	[28,29,30]	[28,31,32]	[33]
Density (kg/m^3^)	1170	1310	1800
Tensile strength, ultimate (MPa)	71	111	4900
Tensile modulus (GPa)	2.5	4.3	230
Elongation at break (%)	53.9	13.9	2.1
Mode I fracture toughness, $G_{IC}$ (kJ/m^2^)	0.2	0.5	-

**Table 2 polymers-13-03642-t002:** Quasi-static mechanical properties of T700/PA6.6 (FVF = 52%) and T700/PPS (FVF = 61%) [28,34,35].

Mechanical Properties	Material: T700/PA6.6	T700/PPS
Tensile Young’s modulus (GPa)	65	60
Compressive Young’s modulus (GPa)	69	47
Tensile strength (MPa)	918	852
Compressive strength (MPa)	461	265
In-plane shear modulus (GPa)	3.2	3.3
In-plane shear stress at 5% (MPa)	52	73
Mode I interlaminar fracture toughness, $G_{IC}$ (kJ/m^2^) [36]	1.50	1.75
Mode II interlaminar fracture toughness, $G_{IIC}$ (kJ/m^2^) [28]	1.94	1.41
Translaminar tensile fracture toughness, $G_{Ic}^{T}$ (kJ/m^2^)	235	314

**Table 3 polymers-13-03642-t003:** Summary of the LVI performance of the different composite systems relative to the T700/PA6.6 system.

Impact Energy (J)	Material	Areal Weight (kg/m^2^) ^a^	Energy Absorption per Areal Weight (kJ.m^2^/kg) ^b^	Percentage Difference vs. T700/PA6.6
100	T700/PA6.6	8.158	11.77	-
	T700/PPS	6.807	13.51	+14.8
160	T700/PA6.6	8.158	20.37	-
	T700/PPS	6.807	23.08	+13.3

^a^ Areal weight = weigh of panel / area of panel. ^b^ Energy absorption = area under the force-displacement plot/areal weight.

**Table 4 polymers-13-03642-t004:** Summary of damage areas relative to the T700/PPS system in comparison to T700/PA6.6.

Impact Energy (J)	Material	Relative Damage Area
40	T700/PA6.6	1.0
	T700/PPS	3.2
100	T700/PA6.6	1.0
	T700/PPS	1.8
160	T700/PA6.6	1.0
	T700/PPS	1.2

**Table 5 polymers-13-03642-t005:** Computational time vs. element size with respect to the maximum load.

Element Size (mm)	No. of Elements on Panel	Computational Time (Hours)	Maximum Load (kN)
0.5	240,000	34.5	9.6
1.0	60,000	3.0	9.7
2.0	15,000	0.6	8.7
5.0	2400	0.02	9.0

**Table 6 polymers-13-03642-t006:** Parameters used to describe the mechanical properties of the T700/PA6.6 and T700/PPS CFRTP composite [28,36] and the contact surface.

Decoupled In-Plane Mechanical Properties	T700/PA6.6	T700/PPS
Longitudinal Young’s modulus, in the fibre direction, $E_{a}$ (GPa)	129	118
Transverse Young’s modulus, $E_{b}$ (GPa)	5.0	7.9
Shear modulus, $G_{ab}$ (GPa)	3.2	3.3
Longitudinal tensile strength, $X_{t}$ (MPa)	2400	2100
Longitudinal compressive strength, $X_{c}$ (MPa)	1500	1300
Transverse tensile strength, $Y_{t}$ (MPa)	156	261
Transverse compressive strength, $Y_{c}$ (MPa)	189	313
Shear strength, $S_{ab}$ (MPa)	110	136
**In-plane fracture toughness**		
Translaminar fracture toughness in compression, $G_{Xc}$ (kJ/m^2^)	350	460
Translaminar fracture toughness in tension, $G_{Xt}$ (kJ/m^2^)	470	620
Transverse fracture toughness in compression, $G_{Yc}$ (kJ/m^2^)	4.0	4.1
Transverse fracture toughness in tension, $G_{Yt}$ (kJ/m^2^)	4.0	4.1
**Contact (cohesive) surface properties**		
Normal failure stress, NFLS (MPa)	60	60
Shear failure stress, SFLS (MPa)	120	120
Exponent in the damage model XMU, PARAM	1.8	1.8
Mode I interlaminar fracture toughness, $G_{IC}$ (kJ/m^2^) [36]	1.50	1.75
Mode II interlaminar fracture toughness, $G_{IIC}$ (kJ/m^2^) [28]	1.94	1.41

**Table 7 polymers-13-03642-t007:** Properties of the contact surface.

Contact (Cohesive) Surface Properties	T700/PA6.6	T700/PPS
Normal failure stress, NFLS(MPa)	60	60
Shear failure stress, SFLS(MPa)	120	120
Exponent in the damage model XMU, PARAM	1.8	1.8
Mode I interlaminar fracture toughness, $G_{IC}$ (kJ/m^2^) [36]	1.50	1.75
Mode II interlaminar fracture toughness, $G_{IIC}$ (kJ/m^2^) [28]	1.94	1.41

**Table 8 polymers-13-03642-t008:** Impact energy and the corresponding impactor’s initial velocity.

Impact Energy (J)	Prescribed Velocity (m/s)
40	5.77
100	8.77
160	11.21

**Table 9 polymers-13-03642-t009:** Comparison of numerical and experimental results.

Material	Impact Energy (J)	Calculated Area (mm^2^)		Discrepancy vs. Experiment (%)
		Experimental	FE Model	
T700/PA6.6	40	683	782	+15
	100	1502	1281	−15
	160	2705	1753	−35
T700/PPS	40	1781	1019	−43
	100	6686	8545	+28
	160	7098	10,266	+45

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
