# Peer review of "Experimental and Numerical Analysis of Low-Velocity Impact of Carbon Fibre-Based Non-Crimp Fabric Reinforced Thermoplastic Composites"

_polymers, 2021, doi:10.3390/polym13213642_

Round 1

Reviewer 1 Report

The title is not correct.

It should be modified as:

Experimental and numerical analysis of low-velocity impact of 
carbon fibre based non-crimp fabric reinforced thermoplastic composites.

The Figure 4 is too low with resolution. Each segment must be enhanced for clarity and presented separately with color codes/levels.

In Table 3, the designation of footnotes as power of units e.g. ((kJ.m2/kg)2 are misleading. The footnotes be better designated by alphabets and not numeric numbers.

Figure 11 is not necessary.

Conclusion is too long and repeats information from discussion. It should be reorganized as generic inferences derived from the experiments and the observations.

Author Response

Thank you for your feedback. I have now made the following changes:

  1. I have reworded the title
  2. I have inserted the highest possible image quality
  3.  I have changed the footnote designation to alphabets instead of numbers
  4. I understand that figure 11 may not entirely be necessary, but I thought that it would give the reader a visual representation of the element size vs. quarter of the panel size. It simply could be useful to some.
  5. I have removed several sentences and rephrased some from the conclusion to enhance clarity.

Reviewer 2 Report

The paper titled “Experimental and numerical analysis of low-velocity impact of 2 non-crimp fabric carbon fibre reinforced thermoplastic compo-3 sites” is relevant to the journal and falls within the journal's scope, and provides valuable information to the readers. Adequate experimental techniques and results that support the findings are discussed in detail. I have a few minor suggestions that must be incorporated, and the manuscript may please be revised

  1. The authors should rewrite the abstract according to the following flow
    1. Problem Statement (1-2 lines)
    2. Introduction (2-3 lines)
    3. Methodology (3-4 lines)
    4. Salient findings (3-4 lines)
    5. Conclusions (1-2 lines)
  2. Page 6, line 206 author should explain why they choose three energy levels (40, 100, and 160J) for better understanding.
  3. The authors need to give references to equations 1 and 2.
  4. Some grammatic errors are observed in the manuscript that needs to be addressed.

Author Response

Thank you for your feedback. I have since made several changes and updated the paper based on your comments. The summary is as follows:

  1. I have added a few lines and rephrased several sentences in the abstract to accommodate your comments.
  2. I have indicated in the abstract that the energies were chosen to achieve different levels of penetrability; I have also indicated this on line 206 previously. Now, I have added another sentence indicating that the energy levels were done based on some rough calculation and our group's previous experience in impacting 4mm laminated composite panels.
  3. I have now referenced equations 1 and 2
  4. I have also made several other changes overall, so, I may have already addressed the grammatical errors that you have observed
